# *Gh**TBL34* Is Associated with Verticillium Wilt Resistance in Cotton

**DOI:** 10.3390/ijms22179115

**Published:** 2021-08-24

**Authors:** Yunlei Zhao, Huijuan Jing, Pei Zhao, Wei Chen, Xuelin Li, Xiaohui Sang, Jianhua Lu, Hongmei Wang

**Affiliations:** 1State Key Laboratory of Cotton Biology, Institute of Cotton Research, Chinese Academy of Agricultural Sciences, Anyang 455000, China; jing_huijuan@163.com (H.J.); zhaopei@caas.cn (P.Z.); chenwei01@caas.cn (W.C.); sangxiaohui@caas.cn (X.S.); lujianhua@caas.cn (J.L.); 2Zhengzhou Research Base, State Key Laboratory of Cotton Biology, Zhengzhou University, Zhengzhou 450000, China; 3Agricultural College, Henan University of Science and Technology, Luoyang 471000, China; xuelin@haust.edu.cn

**Keywords:** trichomes birefringence-like protein, Verticillium wilt resistance, cotton

## Abstract

Verticillium wilt (VW) is a typical fungal disease affecting the yield and quality of cotton. The Trichome Birefringence-Like protein (TBL) is an acetyltransferase involved in the acetylation process of cell wall polysaccharides. Up to now, there are no reports on whether the *TBL* gene is related to disease resistance in cotton. In this study, we cloned a cotton *TBL34* gene located in the confidence interval of a major VW resistance quantitative trait loci and demonstrated its relationship with VW resistance in cotton. Analyzing the sequence variations in resistant and susceptible accessions detected two elite alleles *GhTBL34-2* and *GhTBL34-3*, mainly presented in resistant cotton lines whose disease index was significantly lower than that of susceptible lines carrying the allele *GhTBL34-1*. Comparing the TBL34 protein sequences showed that two amino acid differences in the TBL (PMR5N) domain changed the susceptible allele GhTBL34-1 into the resistant allele GhTBL34-2 (GhTBL34-3). Expression analysis showed that the *TBL34* was obviously up-regulated by infection of *Verticillium dahliae* and exogenous treatment of ethylene (ET), and salicylic acid (SA) and jasmonate (JA) in cotton. VIGS experiments demonstrated that silencing of *TBL34* reduced VW resistance in cotton. We deduced that the *TBL34* gene mediating acetylation of cell wall polysaccharides might be involved in the regulation of resistance to VW in cotton.

## 1. Introduction

Cotton (*Gossypium* spp.) is an important economic crop in the world and provides the most important natural fiber for the textile industry. Verticillium wilt (VW) is a typical fungal disease caused by *Verticillium dahliae* Kleb, which seriously affects the yield and quality of cotton [1]. The pathogen of VW usually exists in the soil in the form of microsclerotia and invades the xylem and vascular tissues from the roots of the host plant, then expanding to the above-ground parts and leading to leaf chlorosis, necrosis or wilting, leaf and boll abscission, and plant death [2]. It is extremely difficult to control cotton VW because of the persistence of the microsclerotia, the broad host range of the pathogen, scarcity of resistance in Upland cotton, and the unavailability of fungicides to kill the pathogens once they enter the xylem [1]. By conducting introgression of resistance genes from *Gossypium barbadense* L., one domesticated allotetraploid specie with high VW resistance, to Upland cotton or gene pyramiding from different sources of resistance in Upland cotton, breeders had created some VW-resistant Upland cotton cultivars to effectively control VW in cotton [3].

Two views contribute to the pathogenic mechanism of *Verticillium dahliae*. One believes that *V. dahliae* produces a large number of hyphae and conidia after invading the duct tissue cells of cotton, and stimulates the parenchyma cells to produce colloidal substances and intrusion bodies that block the ducts and cause difficulty in transporting water [4]; the other considers that *V. dahliae* produces some phytotoxic metabolites inducing phytoalexin formation and triggering wilt symptoms associated with disease development [5]. On the other hand, the plant itself also has a disease-resistant mechanism adapted to stressful conditions, such as plant cell wall-mediated immunity [6,7]. It was found that there are differences in the vascular structure of resistant and susceptible cotton varieties [8]. As the first barrier of the pathogen entry into host plants, cell walls play an important role and can effectively resist the invasion of pathogens [9]. The complex structure and function of the cell wall require a large number of enzymes and related proteins, so there are many genes related to the synthesis and dynamic transformation of the cell wall [10]. For example, the *bc3*, *bc6*, *bc12,* and *bc14* genes of Arabidopsis mutants were related to the synthesis of cell wall polysaccharides, and the growth of the mutant plants containing the above genes was blocked and showed weakened disease resistance, which proves the importance of polysaccharides in plant growth and disease resistance [11,12,13].

There are a large number of polysaccharide acetylation modifications in the cell wall. It was shown that the modification phenomenon in the plant cell wall has a certain effect on resisting the invasion of pathogens [14]. The Reduced Wall Acetylation (RWA) gene *RWA2* is involved in the acetylation process of cell wall xylan. Compared with the wild type, the Arabidopsis mutant rwa2 has a reduced level of polysaccharide oxygen acetylation, and its resistance to *Botrytis cinerea* is different [15]. In addition to the RWA protein, the trichome birefringence (TBR) protein family and the TBR-like protein family (Trichome Briefringence-Like, TBL) are also involved in the acetylation process of cell wall polysaccharides [16]. Although the acetylation modification of cell wall polysaccharides is widespread, the molecular mechanism of how to process and regulate the acetylation of cell wall polysaccharides did not make a breakthrough until recently. Based on existing research, Schultink et al. [17] designed a hypothetical model of cell wall polysaccharide acetylation. It is speculated that RWA protein plays a role in regulating the acetylation modification of polysaccharides; TBL protein may be an acetyltransferase, which determines the specificity of acetylation modification; Altered Xyloglucan 9 (AXY9) may be an acetyl donor with unknown function [17]. Although the functions of different proteins have been speculated, the specific mechanism of the acetylation is still unclear.

Currently, there are many domains of unknown function protein families (DUFs) in the protein family database. Some of these proteins are involved in the acetylation of cell wall polysaccharides, encode related acetyltransferases, and participate in the regulation of plant cell wall development [18]. For example, the xylan acetyltransferase ESK1/TBL29 in the Arabidopsis DUF231 family has been confirmed to be related to the acetylation of cell wall polysaccharides [19]. There are 46 members of the TBL gene family in Arabidopsis, and the family has two conserved domains of TBL (PMR5N, PF14416) and DUF231 (PC-Esterase, PF13839), containing glycine-aspartic acid-serine (Gly-Asp-Ser, GDS) and aspartic acid-x-x-histidine (Asp-x-x-His, DxxH) conserved motifs, respectively. Genes in this family can specifically recognize acetylation substrates and participate in the acetylation process of cell wall polysaccharides, some also protecting plants from pathogens and environmental stress [16,20]. Studies showed that mutations of *AXY4/TBL27* and *AXY4L/TBL22* made the xyloglucan unable to undergo acetylation modification in the cell wall, verifying their relationship with acetylation [14]. ESKIMO1 (ESK1)/TBL29 is not only involved in the process of acetylation modification, but also a negative regulatory gene that allows cold stress, which is related to plant resistance to cold and frost resistance [21]. PMR5/TBL44 is more sensitive to the harm of metals, especially aluminum, and the pectin content from the mutant increases, which may also participate in the formation of pectin [22,23]. The absence of TBL34 and TBL35 in Arabidopsis will result in severe atrophy of the xylem vessel, changes in the secondary wall structure, and extremely slow plant growth, which significantly reduces the plant height of Arabidopsis and weakens the strength of the stalk [24]. The homologous sequences *OsTBL1* and *OsTBL2* of Arabidopsis *TBL34* are not only related to the acetylation of rice xylan, but also affect the resistance to bacterial blight in rice. Compared with wild type, mutants lacking *OsTBL1* and *OsTBL2* genes are more susceptible to disease [20]. These studies show that the acetylation of cell wall polysaccharides, especially xylan, is essential for plant growth and development, disease resistance, and stress resistance. However, there are only a few studies on whether this type of gene is related to disease resistance in cotton.

In our previous study, we detected a cotton *TBL34* gene (*Ghir_A01G022360*), which is homolog of Arabidopsis *TBL34* [24] and located in the confidence interval of a major VW resistance QLT, *qRDI-A01-1* [3], and showing significant induction upon *V. dahliae* inoculation based on the transcriptome sequencing from a disease-resistant inbred cotton line inoculated with *V. dahliae* [25]. In this study, the genomic sequences of the cotton *TBL34* in resistant and susceptible accessions were analyzed to detect the elite allele related to VW resistance in cotton. The gene structure, expression pattern, and primary biological function were estimated to demonstrate that cotton *TBL34* is responsible for VW resistance.

## 2. Results

### 2.1. Analysis of Sequence Variation of GhTBL34 and Detection of Elite Alleles Related to VW Resistance in Cotton

Genome sequences of *GhTBL34* were amplified from 16 cotton lines with the extreme phenotype in VW resistance using primers designed based on *Ghir_A01G022360*. The alignment results of 16 sequences indicated 1 InDel and 18 SNPs, resulting in 3 alleles with the length of 2126 bp (*GhTBL34-1*), 2129 bp (*GhTBL34-2*), and 2134 bp (*GhTBL34-3*), respectively (Figure 1). *GhTBL34-1* mainly appears in susceptible cotton lines such as JM11, while *GhTBL34-2* and *GhTBL34-3* mainly present in resistant cotton lines such as ZZM2 and Acrot-1 (Figure 1 and Figure 2). T test showed that disease index of resistant lines carrying the alleles *GhTBL34-2* and *GhTBL34-3* was significantly lower than that of susceptible lines carrying the allele *GhTBL34-1* (Figure 3), implying that *GhTBL34-2* and *GhTBL34-3* were elite alleles that could significantly improve VW resistance in cotton.

The full-length complementary DNA (cDNA) of *GhTBL34* was cloned and compared with the corresponding genomic sequence to determine the gene structure. Structure analysis showed that *GhTBL34* contained five exons and four introns (Figure 4). Of the 19 sequence variations, 6 SNPs were located in exons, 12 SNPs, and 1 InDel in introns (Figure 4, Appendix A).

The predicted GhTBL34 protein is 323 amino acids long and contains two domains, TBL (PMR5N) (PF14416, 1th–28th amino acid of GhTBL34) and PC-Esterase (PF13839, 29th–319th amino acid of GhTBL34), which were highly conserved among different plant species, all of which contained the *TBL* gene–specific GDS motif (50th–52th amino acid of GhTBL34) and DXXHWCLPGXXDXWN motif (299th–313th amino acid of GhTBL34) (Figure 5). By comparing the TBL34 protein sequences from GhTBL34-1, GhTBL34-2, GhTBL34-3, GbTBL34, and TBL34 of other plant species, it was showed that GhTBL34-2 and GhTBL34-3 had the same amino acid sequences, and two amino acid differences were found to be located in the TBL (PMR5N) domain (Figure 5), one positioned in the 14th amino acid of GhTBL34, which changed the asparagine (Asn, N) of GhTBL34-1 into the aspartic acid (Asp, D), the other positioned in the 16th amino acid of GhTBL34, which changed asparagine (Asn, N) of GhTBL34-1 into the lysine (Lys, K) (Figure 5). Interestingly, the protein sequences of the resistant allele GhTBL34-2 had higher holomogy with those of GbTBL34 from the resistant cultivar Hai7124, both of which shared the same amino acid sequences of the TBL (PMR5N) domain, while the protein sequences of the susceptible allele GhTBL34-1 in the TBL (PMR5N) domain was more similar with those from *G. arboreum* (Figure 5), which had a weak VW resistance. It was deduced that the amino acid differences in the TBL (PMR5N) domain led to the change of VW resistance.

### 2.2. Expression Pattern of Cotton TBL34

Transcript levels of cotton *TBL34* relative to ACT14 in different tissues (roots, stems and leaves) were analyzed in the resistant cultivar ZZM2 and the susceptible cultivar JM11 by qRT-PCR. It was shown that *TBL34* expressed in all of the tested tissues, but was highly expressed in roots and little in leaves (Figure 6A). The transcript levels of *TBL34* in ZZM2 were higher than those in JM11 in each tissue, and significant differences were detected between the resistant cultivar ZZM2 and the susceptible cultivar JM11 in the stem tissue (Figure 6A).

The relative expression levels of *TBL34* in the resistant and susceptible cultivars were determined after inoculation with *V. dahliae*. Expression of *TBL34* in the stem tissue of ZZM2 and JM11was up-regulated at six hours post inoculation (hpi), and began to decrease at 12 hpi (Figure 6B). However, at 12 hpi, *TBL34* transcript levels were significantly higher in ZZM2 than that in JM11 (Figure 6B). In the root tissues, *TBL34* transcript levels in two resistant cultivars, ZZM2 and Hai7124, were significantly higher than those in the susceptible cultivar JM11 at 6, 24, and 48 h after inoculation (Figure 6C). So, it was deduced that the up-regulated expression of *TBL34* in the resistant cultivars after inoculation with *V. dahliae* might contribute to VW resistance in cotton.

The effect of plant hormone (salicylic acid (SA), Ethylene (ET), and Jasmonic acid (MeJA)) treatments was estimated using qRT-PCR. It was showed that *TBL34* expression was up-regulated under exogenous treatment of ET, SA, and MeJA (Figure 6D). The ET treatment had a significant effect on *TBL34* expression at each time point after inoculation, while the SA only at 6 and 48 h after inoculation and the MeJA only at 6 and 24 h after inoculation (Figure 6D).

### 2.3. Silencing of TBL34 Reduced VW Resistance in Cotton

The resistant cultivar Hai7124 was used to study the function of cotton *TBL34* using a tobacco rattle virus (TRV)-based VIGS system. A *GbTBL34* fragment was amplified using the appropriate primer and integrated into pTRV2 vector to generate *TBL34*-knockdown cotton lines. When plants infiltrated with pTRV2:*GhPDS* showed bleaching in newly emerged leaves (Figure 7A), silencing efficiency was assessed using qRT-PCR, which exhibited lower expression levels of the *GbTBL34* gene in infiltrated pTRV2:*GbTBL34* plants than the control (Figure 7B). We inoculated these plants by dipping the roots of cotton seedlings in *V. dahliae* Vd080 spore suspensions when the first true leaf was unfolded. After three weeks, the control plants seldom exhibited leaf chlorosis and wilting, while the infiltrated pTRV2:*GbTBL34* plants showed significant symptoms of VW, regardless of planting the control and infiltrated pTRV2:*GbTBL34* plants separately or in the same nutrition pots (Figure 7C). The disease index (DI) values of infiltrated pTRV2:*GbTBL34* plants and control plants at 15 and 30 days after inoculation were analyzed. It was shown that the DI value of the infiltrated pTRV2:*GbTBL34* plants was significantly higher than that of the control plants at both 15 and 30 days after inoculation (Figure 7D). The stems of the control and infiltrated pTRV2:*GbTBL34* plants at 30 days after inoculation were cut through the middle of the stem vascular bundle and used for recovery experiments by analyzing the level of *V. dahliae* colonization. The results showed that fungi colonized in the stems from the infiltrated pTRV2:*GbTBL34* plants were more than those from the control (Figure 7E), implying that the infiltrated pTRV2:*GbTBL34* plants suffered more severe disease. Trypan blue dye experiments were conducted to assay the cell state of leaves from the control and infiltrated pTRV2:*GbTBL34* plants at 30 days after inoculation. The leaves of infiltrated pTRV2:*GbTBL34* plants had larger and darker blue areas, gathered around veins, than those of the control plants (Figure 7F), indicating that the infiltrated pTRV2:*GbTBL34* plants had more dead cells of leaves than the control plants after *V. dahliae* inoculation. The above results suggest that silencing of *TBL34* weakened VW resistance in cotton.

## 3. Discussion

VW, as a typical soil-borne fungal disease, is an important disease that affects the yield and quality of cotton worldwide. As the first barrier of the pathogen entry into host plants, cell walls play an important role of preventing the invasion of pathogens [9]. Plants often incorporate acetyl substituents into cell wall polymers using at least three groups of proteins, RWAs, TBL, and AXY9, to guarantee the proper roles of cell wall [20]. The TBL proteins have been identified as a xylan acetyltransferase in Arabidopsis and Rice [20,24]. In Arabidopsis, TBL34 and TBL35 were demonstrated to mediate 3-O-monoacetylation and 2,3-di-O-acetylation of xylan, which affected the secondary wall structure [24]. In rice, it was shown that OsTBL1 transfers acetate to both 2-O and 3-O sites of xylosyl residues, and *ostbl1* and *tbl1 tbl2* displayed susceptibility to rice blight disease, indicating that this xylan modification is required for pathogen resistance [20]. In this study, we identified a cotton *TBL34* gene that encodes a protein with TBL domain and PC-Esterase domain containing the *TBL* gene-specific GDS motif and DXXHWCLPGXXDXWN motif, which is homolog of Arabidopsis TBL34 [20]. We demonstrated the roles of cotton *TBL34* in defense responses to VW by analyzing the gDNA sequence variation of *GhTBL34*, studying the expression differences of cotton *TBL34* and conducting the VIGS experiments. Our results were in accordance with the previous studies, which concluded that members of the TBL protein family had been shown to impact pathogen resistance [14].

The gDNA sequences of *GhTBL34* was cloned from the resistant and susceptible cultivars. Analysis of sequence variation detected three alleles showing significant differentiations in resistant and susceptible accessions (Figure 3). The allele *GhTBL34-1* mainly appears in susceptible cotton lines, while the alleles *GhTBL34-2* and *GhTBL34-3* mainly present in resistant cotton lines, and significant differences of disease index between resistant lines carrying *GhTBL34-2* and *GhTBL34-3* and susceptible lines carrying *GhTBL34-1,* suggesting that *GhTBL34-2* and *GhTBL34-3* are elite alleles that could significantly improve VW resistance in cotton. By comparing the TBL34 protein sequences, it was found that the 14th and 16th asparagines (Asn, N) of GhTBL34-1, located in the TBL (PMR5N) domain, changed into the aspartic acid (Asp, D) and the lysine (Lys, K) of GhTBL34-2 and GhTBL34-3, respectively, suggesting that the 14th and 16th amino acid differences of GhTBL34 in the TBL (PMR5N) domain led to the change of VW resistance. We deduced that the *GhTBL34*, being located in the confidence interval of a major VW resistance QLT, *qRDI-A01-1* [3], might be a candidate for VW resistance in cotton. Analysis of gene expression showed that the transcript levels of *TBL34* in resistant cultivar were significantly higher than those in the susceptible cultivar before and after inoculation with *V. dahliae* (Figure 6A–C), implying that the up-regulated expression of *TBL34* in the resistant cultivars before and after inoculation with *V. dahliae* might contribute to VW resistance in cotton.

It was reported that plant hormones such as SA, JA, and ET played a vital role in plant defense against various pathogens [26]. In our research, *TBL34* was obviously up-regulated by VW inoculation and exogenous treatment of ET, SA, and JA in cotton. It is generally believed that ET is a plant disease resistance hormones [27], and several genes were rapidly induced during exogenous ET treatment and *V. dahliae* inoculation, including *GbERF1*, *GbERF2,* and *GbERF1-like* [28,29,30]. These genes regulated the expression of downstream resistance proteins and increased the plant disease resistance. We speculated that cotton *TBL34* was regulated by ET, which played a key role in plant defense against VW. As one of the key defense-related hormones, SA plays a major role in establishment of systemic acquired resistance (SAR), a long-lasting and broad-spectrum disease resistance [31]. The positive defense function of SA in cotton defense against *V.dahliae* had been reported by Mo et al. [32]. We deduced that cotton *TBL34*, up-regulated by exogenous treatment of SA, might be correlated with SA signaling and be involved in the SA-mediated defense responses. JA is fatty acid (α-linolenic acid)-derived phytohormone and plays crucial roles in plant defense responses against infection to necrotrophic and (hemi) biotrophic pathogens [33]. It was reported that JA signaling positively regulated the cotton defense to *V. dahliae* infection [31]. In our research, exogenous treatment of JA caused the up-regulated expression of cotton *TBL34*, implying that this gene was correlated with JA signaling. In fact, researchers considered that ET and JA act synergistically against saprotrophic pathogens [34], and the crosstalk between SA and JA/ET had been reported to deal defense responses [35,36,37], which is in accordance with our results that cotton *TBL34* was regulated by ET, SA, and JA and played a key role in plant defense against VW.

We studied the resistance function of cotton *TBL34* using VIGS. By comparing the differences of disease index, trypan blue staining, and *V. dahlia* recovery experiments between infiltrated pTRV2: *GbTBL34* plants and the control, it was shown that silencing of *TBL34* reduced VW resistance in cotton. It was reported that DUF231-containing proteins, such as TBL28, ESK1/TBL29, TBL30, TBL3, TBL31, TBL32, TBL33, TBL34, and TBL35, possessed xylan acetyltransferase activities catalyzing the transfer of acetyl groups from acetyl-CoA onto xylooligomer acceptors albeit with differential specificities [38], and some members have been shown to be related to the freezing tolerance and disease resistance [20]. Therefore, we deduced that the *TBL34* gene may mediate acetylation of xylan and be involved in the regulation of resistance to VW in cotton.

## 4. Materials and Methods

### 4.1. Plant Materials and Growth Conditions

A total of 16 cotton (*Gossypium hirsutum* L.) lines, including the resistant cultivar Zhongzhimian 2 (ZZM2), the susceptible cultivar Jimian 11 (JM11), and another 14 cotton lines with the extreme phenotype in VW resistance, were used in this study (Appendix A). VW resistance of the 16 lines, which had been evaluated in our previous study [3], was listed in Appendix A. Cotton seedlings were grown in the plastic pot filled with solid culture medium (vol/vol, sterile sand: vermiculite: nutritious soil = 1:1:2) in incubators at 25 °C during the day and 20 °C at night, with 60% relative humidity and a 16/8 h light/dark photoperiod. Young leaves of cotton seedlings were harvested and stored at −80 °C for nucleic acid extraction.

### 4.2. Cloning the gDNA and cDNA of TBL34 Gene in Cotton

Plant total DNA and RNA were extracted using Plant Genomic DNA Kit (TIANGEN, Beijing, China) and RNAprep Pure Plant Kit (TIANGEN, Beijing, China), respectively, according to the manufacturer’s instructions. The cDNA was synthesized using a HiScript^®^II Q Select RT SuperMix for qPCR (+gDNA wiper) (Vazyme, Nanjing, China). The gDNA sequences of *TBL34* gene from the 16 cotton lines were amplified using primers designed according to genome sequence information of *Ghir_A01G022360*, as is shown in Appendix A. The cDNA sequences of *TBL34* gene from ZZM2 and JM11 were amplified using primers designed according to mRNA sequence information of *Ghir_A01G022360* (Appendix A). The PCR amplification was performed on a Bio-Rad PCR thermal cycler (C1000) and the procedure consisted of 94 °C for 10 min, 34 cycles of 98 °C for 30 s, 60 °C for 30 s, 72 °C for 90 s/kb, and 72 °C for 10 min. A 20 μL reaction system was used, containing 10 × PCR buffer for KOD-Plus-Neo, 0.2 mmol·L^−1^ dNTPs, 1.5 mmol·L^−1^ MgSO_4_, 0.3 μmol·L^−1^ forward primer and 0.3 μmol·L^−1^ reverse primer, 0.4 U of KOD-Plus-Neo (TOYOBO, Osaka, Japan) and 200 mmol·L^−1^ template. The PCR product was segregated by agarose gel electrophoresis and purified by FastPure Gel DNA Extraction Mini Kit (Vazyme, Nangjing, China). Purified product was cloned into the pEASY-Blunt Zero Cloning vector (TransGen, Beijing, China) and sequenced by GENEWIZ (Suzhou, China). All primers used in this paper are listed in Appendix A and were synthesized by GENEWIZ.

### 4.3. V. dahliae Materials and Inoculation Methods

One moderate pathogenic strain of the defoliating fungus *V. dahliae*, Vd080, was cultured on potato dextrose agar at 25 °C for six days with inverting. Then, conidia were harvested and grown in liquid Czapek’s medium at 25 °C for seven days with shaking. Czapek’s medium comprised 3% sucrose, 0.2% NaNO_3_, 0.131% KH_2_PO_4_, 0.05% KCl, 0.05% MgSO_4_·7 H_2_O, and 0.002% FeSO_4_·7H_2_O (all w/v). Then, the pathogen solution was filtered with four layers of gauze. The conidia concentration was verified by counting conidia using a hemocytometer under a microscope and diluted by sterilized distilled water to approximately 1 × 10^10^ conidia·L^−1^. When the first true leaf of cotton seedlings was unfolded, the plants were inoculated by gently uprooting the roots and dipping them into Vd080 spore suspensions for 5 min and replanted in pots. At 0, 6, 12, 24, and 48 h after infection of *V. dahliae*, the roots, stems and leaves of the plants were separated, frozen in liquid nitrogen, and stored at −80°C for later use. The samples from the cotton seedlings at 0 h after infection of *V. dahliae* acted as the controls to check the gene expression differences of infected plants in different time points after infection of *V. dahliae.* Every sample was performed with at least three biological repeats.

### 4.4. Hormone Processing and Sampling

A total of 1mM of SA, JA, and ET were prepared and sprayed evenly on the leaves of cotton seedlings growing for about three weeks. The control plants were sprayed with pure water. At 0, 6, 12, 24, and 48 h after spraying the hormones, the roots, stems, and leaves of the plants were separated, frozen in liquid nitrogen, and stored at −80 °C for later use. Every sample was performed with at least three biological repeats.

### 4.5. Vector Construction for Virus-Induced Gene Silencing (VIGS) in Cotton and VIGS Experiments

A cotton *TBL34* (*GbTBL34*) fragment from the resistant cultivar Hai7124 was amplified by VIGS-TBL34-F/VIGS-TBL34-R primers as described in previous steps. The *TBL34* fragment was inserted into pYL156 vector, a TRV-based vector used for VIGS through ClonExpress™ II One Step Cloning Kit (Vazyme, Nanjing, China). The pYL156–GhPDS used as a positive control vector was constructed using the same method. The plasmids containing pYL156–GbTBL34, pYL156–GhPDS, pYL156, and pYL192 were transformed into Agrobacterium tumefaciens strain GV3101 using the freeze–thaw method, respectively. For VIGS, Agrobacterium was harvested and injected into two fully expanded cotyledons of cotton seedlings as previously described. The VIGS experiments were performed with at least three biological repeats and for each repeat there were more than ten plants per constructed vector.

### 4.6. Morbidity Situation Analysis

The disease index (DI) was used to measure the morbidity situation of cotton seedlings after *V. dahliae* infection. According to leaf chlorosis symptoms, cotton seedlings were classified into five grades: 0 (healthy plants, no symptoms on leaves), 1 (one or two cotyledons showing symptoms and no symptoms on true leaves), 2 (both cotyledons and one true leaf showing symptoms), 3 (both cotyledons and two true leaves showing symptoms), and 4 (all of the leaves showing symptoms, symptomatic leaves dropped, the apical meristem was necrotic or the plant died) [39]. The DI was calculated using the following formula: DI = Σ (number of infected plant × disease grade)/(total number of infected plant × 4) × 100%.

### 4.7. Trypan Blue Test

The cell dye Trypan was used to stain dead cells blue. Leaves from infected cotton seedlings were dipped in trypan blue dye solution containing 10 mg of trypan blue, 10 mL of 85% lactic acid, 10 mL of glycerol, 10 mL of phenol, and 10 mL of sterile water. Boiling water bath for 2 min, after cooling, decolorization in chloral hydrate (2.5 g/mL) for three days, and finally eluted with sterile water.

### 4.8. qRT-PCR

Tissue-specific expression of *TBL34* and its differential expression patterns in different conditions were investigated using qRT-PCR. SYBR Primix Ex Taq™ II (Tli RNaseH Plus), Bulk (TaKaRa, Dalian, China) were used for qRT-PCR and a 20 μL reaction volume including 10 μL 2 × SYBR Premix Ex Taq II, 2 μL of cDNA template, 0.8 μL of PCR forward primer (10 μmol·L^−1^), 0.8 μL of PCR reverse primer (10 μmol·L^−1^), 0.4 μL of ROX, and 6 μL of sterile water was used. The qRT-PCR was performed on an ABI 7500 qRT-PCR System (Applied Biosystems, Carlsbad, CA, USA). The qRT-PCR procedure consisted of 95 °C for 30 s, 40 cycles of 95 °C for 5 s, and 60 °C for 34 s. The dissociation curves of each reaction were checked and all reactions were performed with three biological replicates. Cotton actin genes were used as an internal control for normalization of expression values. Results of qRT-PCR were calculated by 2^−ΔΔCt^ method (Livak and Schmittgen 2001) and statistical analysis of qRT-PCR results was conducted using DPS software (IBM, USA).

## 5. Conclusions

The gDNA and cDNA of *GhTBL34* were cloned from the resistant and susceptible cotton cultivars. Analysis of sequence variations of *GhTBL34* from 16 cotton lines with the extreme phenotype in VW resistance detected three alleles showing significant differentiations in resistant and susceptible accessions. The alleles *GhTBL34-2* and *GhTBL34-3*, mainly presented in resistant cotton lines whose disease index was significantly lower than that of susceptible lines carrying the allele *GhTBL34-1*, suggesting that *GhTBL34-2* and *GhTBL34-3* are elite alleles, which could significantly improve VW resistance in cotton. Comparing the TBL34 protein sequences showed that the 14th and 16th amino acid differences of GhTBL34 in the TBL (PMR5N) domain led to the change of VW resistance, suggesting that the *GhTBL34* might be a candidate gene for VW resistance in cotton. Expression analysis showed that *TBL34* was obviously up-regulated by VW inoculation and exogenous treatment of ET, SA, and JA in cotton, and the up-regulated expression of *TBL34* in the resistant cultivars before and after inoculation with *V. dahliae* might contribute to VW resistance in cotton. VIGS experiments demonstrated that silencing of *TBL34* reduced VW resistance in cotton. We deduced that the *TBL34* gene may mediate acetylation of xylan and be involved in the regulation of resistance to VW in cotton.

## Figures and Tables

**Figure 1 ijms-22-09115-f001:**
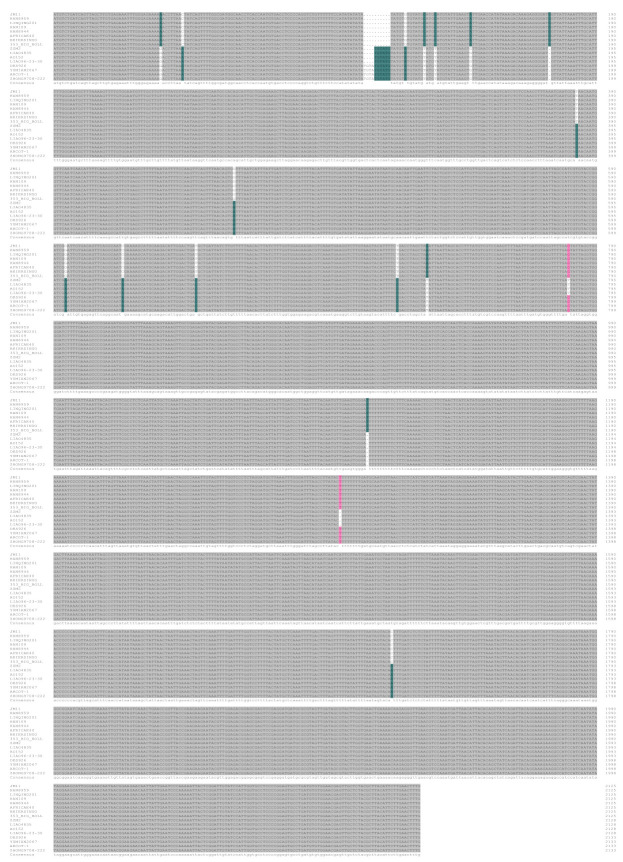
Alignment of *TBL34* gDNA sequences cloned from different cotton accessions.

**Figure 2 ijms-22-09115-f002:**
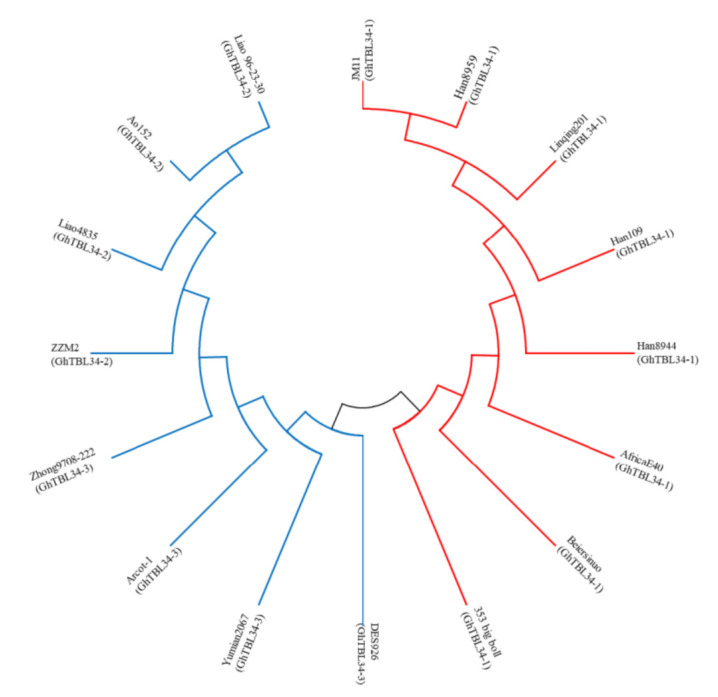
Phylogenetic tree of *TBL34* gDNA from different cotton accessions.

**Figure 3 ijms-22-09115-f003:**
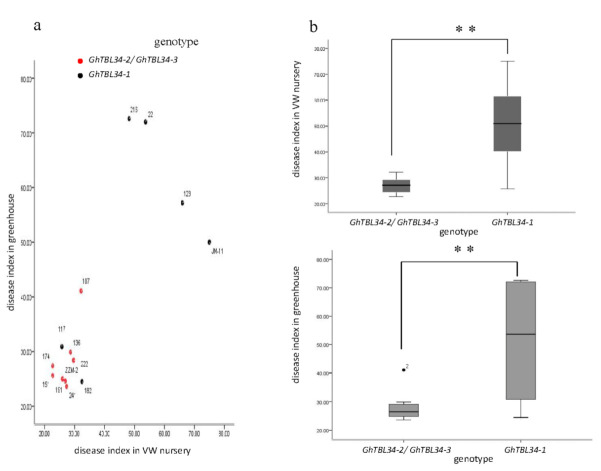
Scatter plot of the distribution of the three alleles in different anti-susceptibility materials (**a**) and the difference of the average disease index in different environments (**b**). **Note:** Both of *GhTBL34-2* and *GhTBL34-3* genotypes are AATTGGAAAATTTTGGNNCCAAAAAACCCCTTIN; *GhTBL34-1* genotype is CCCCAACCGGCCCCAAGGTTGGGGCCTTTTAANN; (**a**) Each number in the figure represents an accession ID, as is shown in Appendix A; (**b**) the middle line indicates the median; the upper and lower limits of the box indicate the upper and lower quartiles of the data; two extension lines indicate the extreme values of the data; the outliers indicate abnormalities value; ** indicated the difference is significant at 0.01 level.2.2. GhTBL34 Gene Structure and Phylogenetic Analysis.

**Figure 4 ijms-22-09115-f004:**
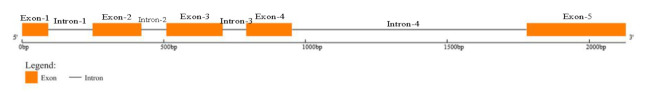
*TBL34* gene structure diagram. Note: Information of sequence variations for each exon and intron refers to Appendix A.

**Figure 5 ijms-22-09115-f005:**
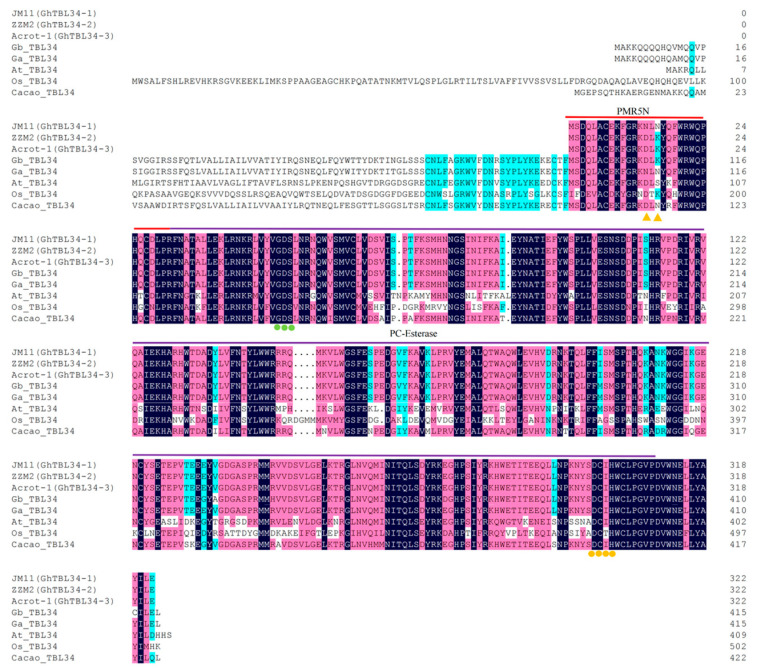
Multiple sequence alignment of GhTBL34-1*,* GhTBL34-2 protein, and TBL34 proteins from other species. Note: The two amino acid differences located in the TBL (PMR5N) domain between GhTBL34-1 and GhTBL34-2/-3 are marked with yellow triangle. The red and purple solid lines represent the conserved domains of PMR5N and PC-Esterase, respectively; the green and yellow dots are the conserved motifs of GDS and DXXH, respectively.

**Figure 6 ijms-22-09115-f006:**
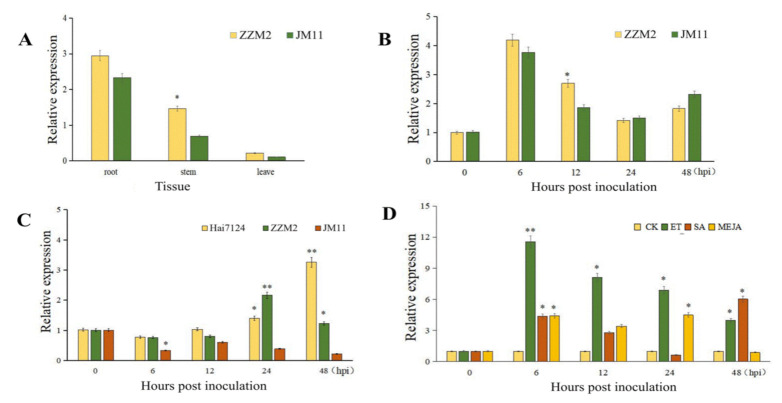
Expression pattern of cotton *TBL34.* (**A**). Tissue-specific expression of *TBL34*; (**B**). Expression levels of the *TBL34* gene in stems of resistant and susceptible cultivars after inoculation with *V. dahliae*; (**C**). Expression levels of the *TBL34* gene in roots of resistant and susceptible cultivars after inoculation with *V. dahliae*; (**D**). Expression levels of the *TBL34* gene in stems of *Gossypium hirsutum* L. treated with different hormones. **Note:** CK = Control; ET = Ethylene; SA = Salicylic acid; MEJA = Jasmonic acid; The significant difference is Student ’s *t* test; * *p* < 0.05; ** *p* < 0.01.

**Figure 7 ijms-22-09115-f007:**
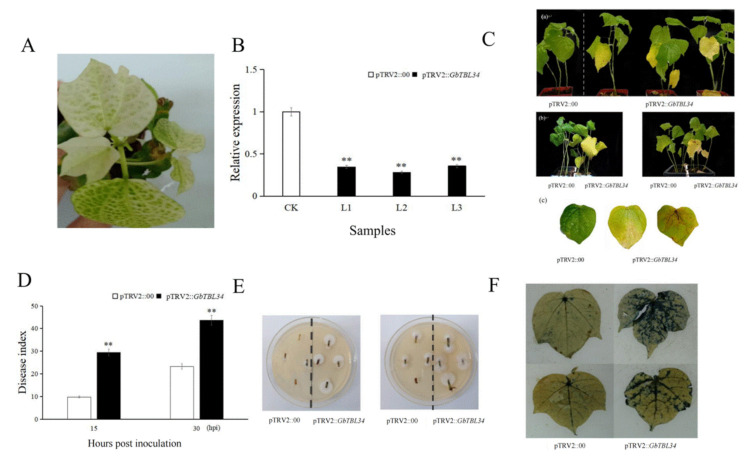
Analysis of the function of cotton *TBL34* using VIGS. (**A**). Phenotype of cotton seedling infiltrated with pTRV2:*GhPDS*; (**B**). Expression levels of *GbTBL34* in silencing plants and control. CK represents the control; L1, L2, and L3 represent the three repeats of the silenced gene plants; (**C**). VIGS phenotype: (**a**) means that the control and infiltrated pTRV2:*GbTBL34* plants were planted in different nutrition pots; (**b**) means that the control and infiltrated pTRV2:*GbTBL34* plants were planted in the same nutrition pots; (**c**) means that the differences of leaves between the control and infiltrated pTRV2:*GbTBL34* plants; (**D**). Disease index of the control and infiltrated pTRV2:*GbTBL34* plants; (**E**). The result of Verticillium dahliae recovery assay; (**F**). Trypan blue staining. **Note:** pTRV2 :: 00 means plants injected with empty vector; pTRV2 :: *GbTBL34* means plants that silence *GbTBL34* gene; The significant difference is Student ’s *t* test; ** *p* < 0.01.

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
