# Peer review of "GhTBL34 Is Associated with Verticillium Wilt Resistance in Cotton"

_ijms, 2021, doi:10.3390/ijms22179115_

Round 1

Reviewer 1 Report

On the basis of their previous QTL mapping results, in this study, the authors characterized one of the candidate genes, GhTBL34, associated with Verticillium wilt (VW) resistance. They compared both genomic and cDNA sequences of the TBL34 gene from cotton cultivars with different level of VW resistance and identified resistant and susceptible alleles. They also compared the expression level of GhTBL34 and its response to Verticillium dahliae (Vd), the causative fungal pathogen of VW, in Upland cotton cultivars resistant or susceptible to Vd. They further down-regulated the expression level of TBL34 in a Vd resistant G. barbadense cultivar using virus-induced gene silencing and found that the level of Vd resistance of the VIGS plants was compromised.

The results presented are of interest to the cotton community and provided a candidate gene for molecular breeding cotton germplasm resistance to Vd. The manuscript can be published after minor revision to correct the followings:

  1. Suggest to change the title to “GhTBL34 is associated with Verticillium wilt resistance in cotton”;
  2. Line 25, delete one of the “of” in “of of cell”;
  3. Line 48, please clarify “plant tissue resistance” (what does that mean?)
  4. Line 90, change “a few” to “few”?
  5. Line 103, change “by” to “based on”;
  6. Line 113, change “12 adding 1” to “12 SNPs and”;
  7. Lines 120-121, where were the “two amino acid differences” found? Between GhTBL34-1 and GhTBL34-2/-3?
  8. Replace the sequences in Fig. 3 with the corresponding alleles, i.e., GhTBL34-1 and GhTBL34-2/-3;
  9. Line 256, change “be decreased” to “decrease”;
  10. Lines 369 and 380, change “Eth” to “ET”.

Author Response

In this updated manuscript, minor revisions have been done according to the reviewer comments, and a list of responses to the reviewer comments has been submitted.

   The revised manuscript incorporating minor revisions, including all text, figures, tables, and any additional files where changes have been made, has been resubmitted.

   The track changes functionality of word processing program have been used to clearly mark all revisions in the revised manuscript.

   The detailed and itemized responses to the comments of the Reviewer 1 are shown in the following file attached.

Reviewer 2 Report

The authors aimed to identify a cotton TBL34 gene and its association with Verticillium wilt resistance in cotton. The study has an appropriate experimental design and suitable methods to carry out the aims. The provided research is interesting and gives some new results related to the aims. However, there are some points that need to be corrected. After revisions, the study can be considered for publication International Journal of Molecular Sciences.

Comments: some examples

L15: Give in full QLT

L23: Give explanation for ET, SA and MeJA.

L26: Use VW instead of Verticillium wilt.

L33: Kleb – not italic

L34: Use VW instead of Verticillium wilt.

L40: G.barbadense – full name needed. Not introduced yet.

L43: The second Verticillium dahliae should be V. dahliae

L51: Space is missing between ’ [9].’ and ’The’.

L56: Delete space between ’ 11, ’ and ’ 12’.

L94-95: V. dahliae

L106: (Figures …

L255 and L260 and L364 and L365 and L422: V. dahliae

L285: Figure 6: Titles of x axis are missing for all figures

L327: Figure 7D: Title of x axis is missing

L366: salicylic acid (SA), jasmonic acid (JA), and ethylene (Eth

L383: Use VW instead of Verticillium wilt.

L392: Please provide a separate conclusion section.

L492: Short form of Applied & Environmental Microbiology is needed.

L496: Gossypium

L544 Short form of Genetics and Molecular Research is needed.

Author Response

In this updated manuscript, minor revisions have been done according to the reviewer comments, and a list of responses to the reviewer comments has been submitted.

   The revised manuscript incorporating minor revisions, including all text, figures, tables, and any additional files where changes have been made, has been resubmitted.

   The track changes functionality of word processing program have been used to clearly mark all revisions in the revised manuscript.

   The detailed and itemized responses to the comments of the Reviewer 2 are shown in the following file attached.

Reviewer 3 Report

A paper dealing with the identification and characterisation of tbl34 gene in cotton is presented. Gene mining, intron/exon organisation, predicted protein analyses was well performed. Similarly, silencing analyses occurred after qPCR evaluation of the analysed gene.

Author Response

Author’s Response to Reviewer 3

   In this updated manuscript, minor revisions have been done according to the reviewer comments, and a list of responses to the reviewer comments has been submitted.

   The revised manuscript incorporating minor revisions, including all text, figures, tables, and any additional files where changes have been made, has been resubmitted.

   The track changes functionality of word processing program have been used to clearly mark all revisions in the revised manuscript.

   The detailed and itemized responses to the comments of the Reviewer 3 are as follow.

Reviewer 3

A paper dealing with the identification and characterisation of tbl34 gene in cotton is presented. Gene mining, intron/exon organisation, predicted protein analyses was well performed. Similarly, silencing analyses occurred after qPCR evaluation of the analysed gene.

Reviewer 4 Report

The manuscript has an interesting result. I have a few comments that will make the manuscript more strong if it has been changed by the author.  

Comments

Page 9 line327: The Verticillium wilt disease causes wilt symptoms, what is showed in the pictures in yellowing only? 
Page 11 line398: Is the cotton seedlings were growing in soil or in liquid media? Descript it?
Page 12 line420: How the spore suspension was prepared, descript it?
Page 12 line 421: How old is the seedling was when it is inoculated?
Page 12 line 421: How is the control treatment was treated?

Author Response

In this updated manuscript, minor revisions have been done according to the reviewer comments, and a list of responses to the reviewer comments has been submitted.

   The revised manuscript incorporating minor revisions, including all text, figures, tables, and any additional files where changes have been made, has been resubmitted.

   The track changes functionality of word processing program have been used to clearly mark all revisions in the revised manuscript.

   The detailed and itemized responses to the comments of the Reviewer 4 are shown in the following file attached.
